# Deep Learning Empowered Structural Health Monitoring and Damage Diagnostics for Structures with Weldment via Decoding Ultrasonic Guided Wave

**DOI:** 10.3390/s22145390

**Published:** 2022-07-19

**Authors:** Zi Zhang, Hong Pan, Xingyu Wang, Zhibin Lin

**Affiliations:** Department of Civil, Construction and Environmental Engineering, North Dakota State University, Fargo, ND 58018, USA; zi.zhang@ndsu.edu (Z.Z.); hong.pan@ndsu.edu (H.P.); xingyu.wang@ndsu.edu (X.W.)

**Keywords:** nondestructive detection, ultrasonic guided wave, convolutional neural network, data-driven approach, structural health monitoring, machine learning

## Abstract

Welding is widely used in the connection of metallic structures, including welded joints in oil/gas metallic pipelines and other structures. The welding process is vulnerable to the inclusion of different types of welding defects, such as lack of penetration and undercut. These defects often initialize early-age cracking and induced corrosion. Moreover, welding-induced defects often accompany other types of mechanical damage, thereby leading to more challenges in damage detection. As such, identification of weldment defects and interaction with other mechanical damages at their early stage is crucial to ensure structural integrity and avoid potential premature failure. The current strategies of damage identification are achieved using ultrasonic guided wave approaches that rely on a change in physical parameters of propagating waves to discriminate as to whether there exist damaged states or not. However, the inherently complex nature of weldment, the complication of damages interactions, and large-scale/long span structural components integrated with structure uncertainties pose great challenges in data interpretation and making an informed decision. Artificial intelligence and machine learning have recently become emerging methods for data fusion, with great potential for structural signal processing through decoding ultrasonic guided waves. Therefore, this study aimed to employ the deep learning method, convolutional neural network (CNN), for better characterization of damage features in terms of welding defect type, severity, locations, and interaction with other damage types. The architecture of the CNN was set up to provide an effective classifier for data representation and data fusion. A total of 16 damage states were designed for training and calibrating the accuracy of the proposed method. The results revealed that the deep learning method enables effectively and automatically extracting features of ultrasonic guided waves and yielding high precise prediction for damage detection of structures with welding defects in complex situations. In addition, the effectiveness and robustness of the proposed methods for structure uncertainties using different embedding materials, and data under noise interference, was also validated and findings demonstrated that the proposed deep learning methods still exhibited a high accuracy at high noise levels.

## 1. Introduction

Welding is one of the major connection methods used for metallic structures [1,2], including connecting oil/gas metallic pipelines [3,4,5]. Due to the nature of the complex welding process on shop and/or construction sites, different types of welding defects are often reported, including incomplete penetration, lack of fusion, cracking, and undercut. As a result, early-age material and structural degradation in terms of cracks and induced corrosion often initiate from these relatively small flaws resulting from weldment. Inspection of weldment for structural health monitoring (SHM) is thus necessary to ensure welding quality during fabrication, construction processes, and later in the in-service stage. 

Ultrasonic guided waves, as a non-destructive testing method in SHM systems, have been widely used for welding quality and assessment in the past two decades [6,7], especially for large-scale pipelines due to their long-range testing capability. For instance, guided waves in torsional mode [8] and longitudinal mode [9] were used for pipe damage detection. Ultrasonic guided waves have been broadly-applied in weldment inspection due to their sensitivity to material discontinuity and structural faults. Ogilvy [10] presented a method of the ultrasonic ray to trace a butt weld by examining the behavior of waves. David et al. [8] confirmed that the lamb waves through selecting different modes and frequency range provided effective detection of welded steel plates. Arone et al. [11] employed the lamb wave to locate defects and used several parameters to identify and characterize different typologies of defects. Many investigations covered a wide range of applications of ultrasonic guided waves in SHM, including characterizing mechanical performance of welded structures [12], identifying defect of shrink fit welded structures [13], quantifying heat affected zone of welded joints [14], and estimating welding quality of spot welding [9]. Clearly, physical features of guided wave signals from different domains, such as amplitude, correlated coefficients, and RMS [8,11], are widely used for damage characterization. In addition, wavelet transform is also a common method for extracting features [12,13]. With increasingly large volumes of data, numerous uncertainties and complex guided wave propagation post great challenges in data process and decision making. Physics-based damage detection and assessment using ultrasonic signals mainly stem from physical features of signals under different domains, such as use of amplitude, correlated coefficients, and RMS [15,16]. Wavelet transform is also a common method for extracting features from signals [17,18]. Due to intrinsic guided waves propagation and scattering, complex modes generated near weldments post great challenges in data classification. In particular, combined effects introduced from other interferences, including measuring noise and inherent structural uncertainty, hinder the applications of conventional physics-based methods. 

Artificial intelligence (AI) and machine learning methods [19,20,21,22,23,24,25,26,27,28] have become an emerging strategy in recent years to solve complex data fusion and information extraction, therefore offering automated ultrasonic signal feature selection and more accurate predictions. Chen et al. [29] proposed an intelligent defect location algorithm with improved accuracy and execution time. Veiga et al. [30] implemented pattern classifiers through artificial neural networks (ANN) for decoding ultrasonic waves for recognition of defect classes. To avoid noise interference, wavelet transform was combined with the ANN algorithm to denoise and identify the location and size of the defects [31]. The multi-class support vector machine (SVM) was used for welding defects classification [32,33], damage orientation, and severity detection [23,34]. As for large volumes of ultrasonic data, numerous uncertainties, complex guided wave propagation in oil/gas pipelines, and feature extraction may cause limitations. Accordingly, deep learning algorithms, such as convolutional neural networks (CNNs), have successfully emerged in SHM systems to fill the gaps in shallow learning. Recent studies demonstrated that CNNs have improved efficiency and accuracy in several challenging tasks including vibration-based structural damage detection and localization in real time [35], vision-based identification of multiple damage types [36,37], and classification of time series [38,39]. 

As illustrated in Figure 1, this study aimed to explore the feasibility of CNN models in assisting damage detection of structures with weldment through encoding ultrasonic guided wave-based signals. The proposed deep learning framework consisted of a dataset collected by the simulation and data augmenting method, training CNN and testing, and predicting the testing data formed by the CNN models. 

## 2. Ultrasonic Guided Waves and Synthetic Signal Generation 

Ultrasonic guided waves generated by piezoelectric transducers are commonly used as nondestructive testing approaches in structural health monitoring for detecting microcracks or any material discontinuity. The principle of the damage detection is based on change in the guided waves when the excited signals encounter defects or other material discontinuity along a pipe. Mechanical damage causes a complex wave scattering in the form of mode conversion, reflection, and transmission. As a result, acquiring the knowledge of guided wave dispersive behavior, mode shapes, and suitable frequency range will make the damage detection susceptible to control. 

### 2.1. Dispersion Curves

Three main modes emerged when guided waves propagated along a hollow cylindrical shell, including longitudinal modes (L modes), torsional modes (T modes), and flexural modes (F modes). The phase velocities and group velocities of the L modes and F modes are the function of frequency and depend on the thickness and diameter of the pipe [40]. L modes are axisymmetric and have no circumferential displacement, which resemble guided waves in plates [41]. T modes are identical to shear horizontal modes in plates. Previous research [42] proved that the longitudinal L (0, 2) mode achieves all pipe wall coverage because of the axisymmetric characteristics. Figure 2 shows the phase speed and group speed of the elastic wave propagating on the steel pipe, using MATLAB package (PCDisp [43]). The example pipe was made from steel material with an outer diameter of 76 mm and a thickness of 4 mm which is same as the model in Section 2.2 and Section 4.1. According to Figure 2, L (0, 2) mode in range 50 to 150 kHz has lower dispersion, higher speed, and lower distortion, and is commonly used in testing. As such, L (0, 2) mode was select to detect damage in this research.

### 2.2. Calibration of the FE Simulation

This section attempted to calibrate the FE model and quantify its effectiveness to capture characteristics of guided waves propagating along a structure. A three-dimensional pipe was established in COMSOL which is commercial finite element (FE) software. The prototype was a 2000.0 mm long steel pipe according to the literature [44], with the outside diameter of 76.0 mm and the inner diameter of 68.0 mm. Two notch-shaped damages were located symmetrically at 1400.0 mm away from the left side. The dimensions of the defects were 2.0 mm in axial width, 3.0 mm depth and 35.0 mm in circumferential length. The material density was 7850.0 kg/m^3^, Youngs’ modulus was 210 GPa, and Poisson’s ratio was 0.28. Free triangular elements available in COMSOL were used to mesh the pipe. To ensure the calculation accuracy and consider the computational efficiency, the maximum element size was 5.0 mm, and the minimum element size was 2.0 mm. In addition, the maximum time increment should be less than 1/20 of the excitation frequency [45]; thus, the simulation time step was defined as 0.1 μs. The boundary of the pipe was free.

As shown in Figure 3, the excitation signal, *D*(*t*), was defined as a 120 kHz 5-cycle sine function operated with a hanning window by the form [44]:(1)$$D\left(t\right)=A\left(1-cos\frac{2\pi f_{c}t}{n}\right)\sin\left(2\pi f_{c}t\right)$$
where *A* is amplitude of the wave, *n* is the number of the period, and *f_c_* is the wave frequency. In COMSOL, the signal was set as a displacement, located on the left side of the pipe, to simulate the effect of the actuator. The model of the pipe is shown in Figure 4.

The excitation waves were introduced into the pipe at left boundary, and the received sensors were located at the same direction. Figure 5 illustrated the signals generated from this study as compared to the those from the literature [44], which exhibited the excitation that traveled through the pipe, and reflected from the defects and the boundary. The first packet of each signal represented the initial wave. Then, reflections of these two signals were similarly located around 6 × 10^−4^ s, suggesting the echoes from the defect, and converted model. The third packet was located around 8 × 10^−4^ s, generating form the right boundary. The wave packets had similar locations and shapes which represented the simulated model was certificated. In the finite element model, the velocity of the wave was 2 m × 2/0.000735 s = 5442 m/s which was close to the data in Figure 2 (5373 m/s) and the velocity (5361 m/s) in the literature [44]. Please note that a different parameter of the excitation can cause the different amplitude of the signals and the multiple reflection of the wave on damage and boundaries can cause the superposition of waves, resulting in an increase of the cycles in the boundary wave packet. The comparison results verified that the simulated model established in this study was proper for the guided waves simulation in pipes; thus, further exploration was based on this simulation to generate data. 

In this section, a steel pipe model was established with the FE method, which was already verified by the literature and dispersion curve. The model in the following research was derived on the basis of this model, which just changed the length, added weldment part and more damage states. 

### 2.3. Characteristics of the Ultrasonic Guided Waves

To further understand the wave characteristics for a structural component with a weldment, we continue the identical pipe as stated in Section 2.2 in the literature, where the pipe size has the outside diameter of 76.0 mm and the inner diameter of 68.0 mm, and the butt weld is assumed at the quarter location and a notch is located at the mid-span of the length. The COMSOL model for this pipe is plotted in Figure 6, including pipes in undamaged state, pipe with a weld, pipe with a crack, and pipe with weld and a crack. As such, the signal characteristics of a pipe with and without weldment was displayed in Figure 7a–d, where for a comparison, mechanical damage, such as cracking, was also included in a pipe. The excited ultrasonic waves that traveled through a pipe with and without weldment echoed when they reached discontinuity due to weldment or mechanical damage and boundary under different scenarios. To clearly see the reflection packet, the envelope of the wave was plotted with a red line as shown in Figure 7. As illustrated in Figure 7a, an intact pipe only contained two main packets, where the first signal packet presented the excitation and the second packet collected represented the echoed wave from the boundary. With the wave speed and travel time, one could accurately determine the length of the pipe if needed. By contrast, Figure 7b–d reveal enriched echoed waves within and outside of the boundary echo when a pipe experience certain structural nonlinearity or discontinuity. 

Figure 7b shows the resulting waves of a pipe with a girth weld. Moreover two dominant packets as identical to the intact pipe shown in Figure 7a, one could also observe low-amplitude oscillations experienced within and outside of the boundary echo due to the weldment reflection. Such characteristics is often used for data classification. 

Figure 7c–d shows the feature of the guided wave signals associated with varying types of mechanical damage or combined weldment with damage. These waves exhibit apparently non-stationary and nonlinear behavior. Figure 7c–d revealed that with the more complicated scenarios, including mechanical damage or combined weldment with damage, the received signals came from the complicated scattering, reflection, and wave interaction.

Physics-based methods are based on physical characteristics of the guided waves that are often interpreted under frequency-, time-, or time-frequency-domains. In a time domain, physics-based features play an important role in ultrasonic wave feature extraction. Amplitude, energy, and the correlation coefficient are three features which can represent the wave characteristic. The amplitude was obtained by the peak value of the damage wave packet. The energy calculated by the root mean square of wave (RMS) in the damage part were defined as
(2)$$\mathrm{RMS}=\sqrt{\frac{1}{n}\overset{n}{\underset{i=1}{\sum }}e_{i}^{2}}$$
where *n* is the number of data points and *e_i_* is the signal. The correlation coefficient under the damage state was used to compare with that of the health state. In frequency domain, the amplitude was extracted as the features. 

As stated in the literature, amplitude, frequency, or RMS that often result in good classification for signal characteristics may perform poorly, for example, when received signals after coupled scattering and reflection/refraction behaved highly nonlinearly, with high level of noise. Despite great efforts in physics-based methods on damage detection and classification, these methods could be hindered by high variance of structures due to uncertainty, measure noise, and other interferences. Thus, the data-driven methods, such as deep learning, were introduced to tackle this issue, as shown in Section 3.

## 3. Deep Learning-Based Damage Diagnostics

### 3.1. Framework of the Deep Learning for Damage Diagnostics

The convolutional neural network was employed to decode guided wave signals to identify potential damages experienced in a pipe. As illustrated in Figure 8, the learning framework consisted of three main parts, including dataset collected by the simulation and data augmenting method, training CNN models by training set and modifying by validation set, and predicting the testing data formed by CNN models. Specifically, the guided wave dataset simulated by the finite element method was enlarged by adding Gaussian white noise with different SNRs. The classification label was defined by the state that each guided wave belongs to. Then data partition was involved in this process, which 60% of the data for training, 20% for the validation and 20% used to test. The prepared training data were input into the CNN model to establish a classifier for identify the damage state of each signal. The CNN model was modified when the classifier operated on validation data. Training and validating were necessary parts for the CNN model to build and improve the sensitivity of feature learning and the performance of classification. Finally, the testing set was classified by the trained CNN model. Therefore, the damage state of each guided wave was predicted by CNN. 

### 3.2. CNN Architecture 

CNNs are composed of layers containing artificial neurons which are arranged in three dimensions including width, height and depth [46]. In this damage detection study, the input consisted of four guided waves, which is a two-dimensional matrix sized 2000 × 4. Stepping through each layer of the CNN, the matrix is converted into a one-dimensional vector corresponding to the category. Figure 9 illustrated the whole process of CNN. Three main layers were involved in CNN architectures, including the convolution layers, pooling layer, and the fully connected layer.

#### 3.2.1. Convolutional Layer

Convolutional layer is the main part in CNN. Each convolutional block contained the learnable parameters such as weights (filters) and biases. Compared to the input, the width and height of the filter is spatially smaller, but the depth is the same. The feature maps from previous layer were convolved with filters and formed the output through the activation function. The formula of the convolutional layer for a pixel is
(3)$$C_{xy}=\sigma (\overset{h}{\underset{i=1}{\sum }}\sum_{j=1}^{w}\sum_{k=1}^{d}f_{ijk}*X_{x+i,y+j-1,k}+b)$$
where *h* and *w* are the size of filter and *d* is the number of channels of the input, *b* is the bias vector, σ is the activation function, and *f* is the weight matrix. 

Figure 10 showed the convolution operation on a 4 × 4 matrix. A 3 × 3 matrix was acted as the filter which was generated randomly at initial state and updated from the model by the backward propagation algorithm. The stride is equal to 1, thus four subarrays, with the same size as the filter, were generated by sliding along the width and height of the input matrix. Each subarray multiplied with the filter matrix element by element. Then, the output value was formed by the sum of the multiplied values and the bias. The size of the output was smaller than the previous layer because of the stride. 

After the convolution layer, a nonlinear activation function was followed, which was used for introducing non-linearity into the network. Two of the most commonly activation functions used in neural networks are rectifier linear unit (ReLU) and softmax. The function of ReLU and Softmax are expressed as follows: (4)$$f\left(x\right)=\max\left(0,x\right)\;$$
(5)$$f{\left(x\right)}_{i}=\frac{e^{xi}}{\sum_{j=1}^{K}e^{xj}}$$

#### 3.2.2. Pooling Layer

The pooling layer was used to reduce the spatial size of the feature maps to speed up the computation and increase the robustness of feature detection. The most common methods were max pooling and average pooling. In max pooling, the maximum value in filter area was chosen as the output, shown in Figure 11. In detail, the input layer, a 4 × 4 matrix, was operated by the max pooling filter which sized 2 × 2. The stride was designed as 2 which means the next filter should move two elements to the right or down. Then, dimension of the output was declined to 2 × 2, and the value was the maximum elements in the response field. As the average pooling, it took the average value in the filter area.

#### 3.2.3. Fully Connected Layer

The fully connected layer is the one before the output layer of the whole network. In this layer, all the neurons are connected with the features generated by the previous layer. In addition, weights and biases in this layer converted the feature into the correspondent categories. The equation of the output $y^{l}$ is shown
(6)$$y^{l}=\sigma \left(y^{l-1}\times w+b\right)$$
where *σ* is the activation function; ***w*** and ***b*** represent the weights and bias vectors in this layer.

## 4. Case Studies

### 4.1. Design of Scenarios

All the parameters were set as a multiple of the pipe outer diameter, including the length of the pipe, the dimensions, and the location of the defect. The outer and inner diameters, D_out_ and D_in_, were 76 mm and 68 mm, respectively. The length of the pipe was 20 × D_out_, equal to 1520 mm. Figure 12a,b shows the details of the 3D pipe model, where the excitation nodes were set at the left side and four receivers were circumferentially symmetrical located at position A (15mm away from the left side). As illustrated in Figure 12a, weldment was located at the point B and a notch-shaped damage was at point C. The butt weldment was designed as a V-shape cross section, the weld width at the outside and inside surface were 10 mm and 5 mm, respectively, and the thickness was 4 mm higher than that of pipe as shown in Figure 12c. The material welding filler was Ti-6Al-4V. In detail, the density of this material is 4453 kg/m^3^, Young’s modulus of 125.8 GPa, shear modulus of 40 GPa and Poisson’s ration of 0.40. The excitation was 5-cycle sinusoidal signal operated by Hanning window in 100 kHz. The velocity of the wave was close to 5241 m/s which was similar to the value in Figure 2.

The welding manufacturing process can cause impurities in steel structures, resulting in material discontinuity. Four main welding defects were designed in this research, including lack of fusion, cracks, undercut, and lack of penetration, as shown in Figure 13. Contaminated surfaces prevent the welding bead from adhering and fusing with the base metal, resulting in a lack of fusion defects in welds. Cracks occur if the weld metal shrinks as it solidifies. The undercut is the weld that reduced the cross-sectional thickness of the base metal. When the filler material does not fill into the joint, lack of penetration defects appears [47]. Three different severity levels were designed for weld defects, accounting for 10%, 5% and 1% of the circumference, respectively. Clearly, the damage severity in 1% means the arc length of the damage is 1/100 of the circumference of the pipe. 

In addition to the welding part, the pipeline also suffered from load and environmental erosion. Hence, the notch-shaped damages of different positions were considered in the research, representing minute cracks with a width of around 10^−4^ m. The damage depth was 4 mm and the length was 0.1 × D_out_. The damage locations ranged from 0.5 × D_out_ to 10 × D_out_. 

A total of 16 different states were obtained, separated into four cases to investigate the effectiveness of the proposed method. The reference state (State #1) was a pipe with a healthy welding and without damage. In Case 1, the crack location was detected in the pipeline with health welding (State #2–6). Case 2 was also identified as the crack damage location in the pipeline with welding defect interference (State #7–11). The crack, a notch-shaped damage, was located at five different positions, including 0.5 × D_out,_ 1 × D_out,_ 2 × D_out,_ 5 × D_ou_, and 10 × D_out_. Case 3 (State #10, 12–14) was focused on identifying the 4 kinds of welding defects with a certain severity (1% of the circumference). The detection of welding defect severities was carried out in Case 4 (State #10, 15, 16) where had a lack-of-fusion defects in the welding of the pipeline. To consider the noise inevitably mixed in the detection signals, all the states were interfered with by additive white Gaussian noise with the SNR equal to 60 dB to 100 dB. Detailed information is as shown in Table 1. 

The results in Figure 14a–f indicated the details of guided waves propagated and diffracted through the whole span of pipeline during different periods. Taking State #5 as an example, the excitation wave was input from the left side and traveled to the right. When the signal interacted with the weldment at 1 × 10^−4^ s, part of the wave was returned, and the rest was continually moving forward. At 1.8 × 10^−4^ s, the initial wave was interfaced by the notch-shaped damage, and on the other side, the echoed wave was arrived at the left boundary due to the same distance. Meanwhile, the echo was received by the actuators and kept propagating to the right. As the original wave traveled back to the left boundary (at 3.2 × 10^−4^ s), the wave completed an entire propagation in the pipeline, experiencing the reflection from damage and weldment for several times. Therefore, the detection signal included multiple wave packets because the initial wave packet was experienced several times reflections and scattering. 

### 4.2. Data Augmentation through Noise Injection

The data received by sensors were easily contaminated by noise from the environment. As such, appropriate noise added to the collected signal could consider the real situation in the lab. The signal to noise ratio (SNR) represents the ratio of the signal strength to the background noise strength expressed as [48]:(7)$${\mathrm{SNR}}_{dB}=10\log_{10}(\frac{P_{\mathrm{signal}}}{P_{\mathrm{noise}}})\;$$
where $P_{\mathrm{signal}}$ and $P_{\mathrm{noise}}$ are the average power of signal and noise by *dB* scale, respectively. Noise levels ranged from 100 dB to 60 dB have been added in State # 1–16 for forming multiple synthetic signals. Furthermore, classifying signals in different noise levels can check the sensitivity of the deep learning method to the uncertainty caused by noise. Taking the signal in State #5 as an example, Figure 15 shows the waves under different noise levels. Figure 15a shows the initial signal generated by numerical simulation. Clearly, several wave packets represented the echoes from damage or boundary. With the SNR reduced, the signals were highly contaminated. Especially at 60 dB, the features of the signal were covered by noise.

### 4.3. Analyses of the CNN Parameters

Eight-layer CNN architectures derived from LeNet [49] are proposed in this section. In detail, the 8-layer CNN configuration for guided wave-based damage detection included three convolution layers, two max pooling layers, a fullu connected layer, a rectified linear unit (ReLU), and a sigmoid function (Softmax). The important parameters of the CNN architecture are discussed in this section, including the convolutional filter size, number of convolution filters and the parameters of pooling layers. The networks were trained by the guided waves collected from State #1–6 in Table 1 to classify the reference state and 5 different damage location in pipe (damage locations are 0.5 × D_out_, 1 × D_out_, 2 × D_out_, 5 × D_out_, 10 × D_out_). Overall, 3000 data items were enlarged by adding Gaussian white noise with 500 in each state. The size of each data sample was 2000 × 4. The whole samples were split into the training set (60%), validation set (20%), and testing set (20%).

#### 4.3.1. Size of Convolutional Filters

The convolutional filters represent the local features of the input signals. The appropriate size (the width and the height) of the filters can represent the typical features of signals and improve the efficiency of CNN. Since the width of example signals were small, only the heights of the filters in the first two convolutional layer were considered, setting from 5 to 45. Table 2 reported the training results under different noise levels. When SNR was equal to 70 dB, the classification was 100% correct in these five groups. With the noise increase, the accuracy of prediction was drop down. As the size of the first two filters increased, the testing error decreased to 0.99% in Group 3 and then increased under 65 dB. The similar situation was happened at SNR equal to 60 dB, which the lowest testing error was emerged at Group 3 (about 17%). As such, the filter sizes shown in Group 3 were used in convolutional layers for further study. Figure 16 illustrated the weights of the third convolutional layer from Group 1 to 5 at 60 dB. The channel represented the number of filters and the dimension was the length of the filter. After training by two convolutional layers, the third one could better reflect the features learned from the input. Obviously, in five distinct groups, the fluctuations of each feature vector were gradually changed. The weight distributions in Group 1 and 2 were uniform, thus the features were difficult to extract. In Group 3 to 5, the distributions of weights were different; however, Group 3 showed the most notable features which means the architecture made this network well trained.

#### 4.3.2. Number of Convolution Filters

The number of the filters determined the quantity of the feature map generated in this layer, which made it possible to extracted features much more fully. As the network deepened, the length and width of the feature map shrank, indicating that the features extracted in this layer were much more representative. Obviously, too few filters cannot achieve the demand of feature extraction, and too many filters lead to time wasting. The classification results are listed in Table 3. The classification in five groups were compared under different noise levels. Under the lower noise level, the number of filters in each layer had less impact on classification accuracy. However, when noise level increased, the filter number proposed in Group 3 showed better results, achieving 99.17% and 82.83% under 65 dB and 60 dB, respectively. 

#### 4.3.3. Parameters in Pooling Layer

The pooling layer, also known as the down sampling layer, aims to reduce the dimension of the feature maps, while remaining the most representative information in the region. In addition, the pooling layer also prevents overfitting to some extent. Larger size can cause the feature loss; however, smaller size lost the properties of dimensionality reduction. Moreover, the type of pooling layer should be designed before building the entire structure of CNN. Two pooling layers were involved in the CNN architecture. Six distinct combinations including different sizes and types were designed herein attempting to explore the relationship between pooling layer and the accuracy of prediction. Table 4 revealed that the predictions using max pooling were better than that of mean pooling for this dataset. Meanwhile, the parameters of two pooling layers set as Group 3 could achieve better results, at 15 and 8 for the first and second pooling layer, respectively.

Detailed information about the proposed CNN framework is shown in Table 5. The adopted network had an input layer in which the four received signals (sized 2000) were combined and directly sent into the framework. Then the first convolutional layer which the filter size was 25 × 2 and max pooling layer were operated the signals into 393 × 3. Following was the second convolutional layer with 40 filters sized 25 × 3 and max pooling layer which reduced the signal into just one dimension (73 × 1). The third convolutional layer which the filter size changed into 5 × 1, then the full connected layer and softmax layer worked together to map the result into a 4 × 1 matrix. In total, nearly 5000 parameters were generated in this network.

## 5. Results and Discussion

### 5.1. Classification Performance for Case 1 and 2: Detection of Notch-Shaped Damage Locations

#### 5.1.1. Training and Validation Results of Notch-Shaped Damage Locations

The eight-layer CNN proposed in this paper was used to identify the location of damage on the pipeline, which was trained by 3000 records of guided wave signals collected from State #2–6. The pipeline was excited by the 100 kHZ 5-circle sinusoidal signal modulated by Hanning window. Gaussian white noise with five specific levels were added to the signals to investigate the sensitivity and effectiveness of the method interfered by noise. The dataset was partitioned into training, validation, and test set with 60%, 20%, and 20%, respectively. Figure 17 demonstrated feature maps of six categories obtained from the last convolutional layer of the CNN model under the noise levels of 100 dB and 60 dB. The dimension was the length of the output, and each line illustrated the channel of the output. Specifically, the shape of feature maps was more obvious to distinguish the status of the pipeline when SNR was 100 dB. For instance, in the first label, three main wave peaks were revealed after six layer’s training. In the second label, four main wave peaks occurred with different values. Analogously, the six labels substantial display the characteristics of themselves, which become susceptible to passing through the last fully connection layer and obtaining the final probability of each category. However, at 60 dB, the feature maps became messy, representing that the model was hard to extract sensitive features from the detection signals in this condition. Although this was not the final result of the classification, the fully connection layer struggled to achieve reasonable ratios of each category. As a result, the accuracy of CNN classifier would be lower when noise level was higher. 

Figure 18 reveals the training and validation errors under three different noise levels of 80 dB, 70 dB, and 60 dB. As illustrated in Figure 18a, the model extracted the information rapidly at 80 dB, which error rates were changing from 0.85 to 0 after 10 epochs. With the noise level increased, the accuracy of classification decreased. When SNR was 70 dB level (see Figure 18b), the training error was changed slightly at the time of the first 10 epochs, and then the error rate dropped sharply after spending 20 epochs of training from 0.8 to 0.02. At the 23rd epoch, the training error was sent to 0, which took more than 2 computations compared to the situation at 80 dB. Meanwhile, the tendency of the validation error was similar to the training error, except that the rate converged to 0.12 not 0 after 20 epochs training. It means the classification model could not entirely identify the damage location correctly under this noise level. In 60 dB, the classification was deteriorated where the validation value was just close to 0.7 after 100th epoch, demonstrated in Figure 18c. Noise interference had an adverse effect on the CNN classifier for damage location. 

When the weld damage and material discontinuous interacted, the detection by ultrasonic guided waves could be more challenging. Six different damage states (Case 2, State # 7–11) in Table 1 were designed as the training dataset which contained 3000 data points. When the noise level was relatively low, i.e., SNR is higher than 70 dB, CNN model could be fully trained within 10 epochs from the training and validation sets. However, the error rates for the cases at SNR = 70 dB, required higher epochs were declined dramatically in 10 epochs, then, training error was closed to zero and validation error was equal to 0.04 after 15 epochs’ learning, as shown in Figure 19a. With the noise level increased to 60 dB, CNN classifier spent 100 epochs to reduce the training error rates from 0.83 to 0, taking 10 times longer than that of 70 dB, while the validation part only converged to 0.32. The main reason for this should be the input data were hard to identify because of the high noise.

#### 5.1.2. Testing Results of Notch-Shaped Damage Locations

Validating the classification model by testing data, the result of notch-shaped damage position detection in Case 1 (State #1–6) is illustrated in Figure 20. A total of 600 signals were input into the pretrained model, and the prediction result of each data as directly obtained. Obviously, the CNN method trained by a part of the signals could accurately identify the damage location in most situations, which acquired 100% classification at a noise level from 100 dB to 80 dB. Figure 20a showed the confusion matrix of the testing result at 80 dB. However, the special case was happened at 70 dB, which the accuracy of the test decreased dramatically to 92.33%. The data in each category was misled by other categories to some extent. Specifically, two of the signals with damage located at 0.5 D_out_ from the weldment were misclassified as group 1 D and one of them was misled into 5 D_out_. The accuracy of the fifth class was 86%, where eleven of the samples in 1 D group were mistakenly categorized into other groups. It is mainly because the high level of noise that the signals were contaminated and this lead to a high number mislabeled cases. 

As the weld defect was considered to interfere with the identification of damage locations, the testing results are shown in Figure 21. The classification was 100% when SNR was larger than 70 dB. At 70 dB, 95% of the data could be predicted into have the correct labels, in which the mislabeled data mainly occurred in position 5 D and 10 D. The accuracy further dropped to 68%, as the noise level was 60 dB. In the base state, the prediction was 92% and when notch-shaped damage located in 0.5 D, 1 D and 2D, the predictions were 75%, 71%, and 76%, respectively. More misclassifications occurred under this noise level. For instance, 31 data points in 10D group were classified into 5D, and 29 data points were classified into 0.5 D, 1 D and 2D. Due to the multiple relationship between notch-shape defect and weld position, the waves were much easier to interact which made it more difficult to extract features. In addition, noise is one of critical factors affecting the accuracy of prognosis.

The accuracies of classification in each category under different noise level are illustrated in Figure 22. Comparing these two conditions (classifying the notch-shaped damage location without or with the welding defect existed), the predictions could entirely classify the categories under 80 dB, 90 dB, and 100 dB. The misclassifications appeared at 70 dB and 60 dB. Please note that the classifications of damage and undamaged state were 100% in the first situation, as shown in Figure 22a. Clearly, the addition of the welding defect did not affect the efficiency of the classifier, instead, the result is slightly better, because the additional of weld defect increases the complexity of the signal and allows more features to be extracted.

### 5.2. Classification Performance for Case 3: Detection of Weld Defect Types

#### 5.2.1. Training and Validation Results of Weld Defect Types

Welding defect detection is crucial to ensure structural integrity. Several defects, such as lack of fusion, cracks, undercut, and lake of penetration are easily formed during the manufacturing process. Four different welding defects shown in Table 1 (State #10, 12~14) and the reference state (State #1) were composed as the training set when the SNR ranged from 100 dB to 60 dB. The severity of the welding defects was 1% of the circumference and a notch-shaped damage was located at 5D away from the weld. 

Figure 23 shows the accuracy for both training and validation under the noise levels of 70 dB and 60 dB. When SNR is higher than 70 dB, CNN model could lead to higher accuracy. At 70 dB, approximately 50 epochs was cost to convert the ratios into zero. The validation ratio was equal to 0.006 after 200 epochs training. The results worsened at 60 dB, where the error curves were dropped slowly and trembled violently. Specifically, the training error had a slight decline from 0.8 to 0.5 during 40 epochs, then reached zero after learning for 100 epochs. The validation rate was only close to 0.21, i.e., about 21% of data are mislabeled.

#### 5.2.2. Testing Results of Weld Defect Types

The results of the damage type classification are shown in Figure 24. The confusion matrices illustrate the situation of the prediction in each state. In 70 dB, the testing rate was 99.4%, including 3% of the data in defect 4 misled into defect 3. The errors were fewer at 60 dB where the total accuracy was only 75.4%, in which the mainly error was the distinction of defects 3 and 4. The misjudgments in these two categories were higher than 50%, suggesting that the methods were not sensitive to these defects under the noise interference at level of SNR of 60 dB. It was mainly because similarity of echoes of the guided waves for the defects 3 and 4 (undercut and lake of penetration) at outer or inner surfaces as well as the high noise interference led to difficulty in data classification. 

### 5.3. Classification Performance for Case 4: Detection of Damage Severities

#### 5.3.1. Training and Validation Results of Weld Defect Severities

Detecting the severity of damage is also a significant and difficult issue in ultrasonic guided wave-based health monitoring. The severity of welding defects determines the maintenance in pipeline. Therefore, the dataset consisted of three different severities of weld defect (State #10, #15, #16), and one state as the reference (State #1). Each scenario included 500 datasets generated and thus a total of 2000 data points were involved in this learning. The training results of classification by CNN framework are shown in Figure 25 when SNRs were 80 db and 70 dB. The error rates at 80 dB approached zero after the 18 epochs training, while 120 epochs were used training the data error reaching to zero for the noise level of 70 dB, which was more than six times compared to the situation at 80 dB. In addition, the validation was equal to 0.095. 

#### 5.3.2. Testing Results of Weld Defect Severities

Figure 26 reveals the accuracy of the prediction with respect to defect severity. Clearly, the accuracy for the cases under noise levels of 80 dB and 70 dB could reach up to 100% and 88.75%, respectively. The prediction reached to 100% when SNRs higher than 70 dB. However, at 70 dB, 96% of the data labeled 10% was classified as correct, with 4% misled into label 5%. The prediction in the samples of 5% severity was 85% with seven data points misclassified into the case of 10% severity and eight data points were in the case of 1% severity. The lowest accuracy was the defect with a length of 1% in circumference where 26% of data points were misclassified. More errors happened when the length of the defect become smaller and the accuracy dramatically dropped when SNR was at 60 dB. 

Figure 27 shows the accuracy in each label with respect to noise interference, ranging from 100 dB to 60 dB. In general, the data classifier in this study could effectively predict the severity of the defects at most cases. Similar to other observations, the noise levels at 70 dB or 60 dB dramatically reduce the performance of the data classifier. Clearly, the base condition could be much easier to be distinguished with the damage states. The data labeled in 1% severity had the lowest accuracy by 35%, under the noise level of 60 dB.

## 6. Further Discussion on the Effectiveness/Robustness of the Deep Learning Methods 

While we presented the investigation of deep learning-based signal process of the guided waves for structures with weldment and interacting effects of weld defects with mechanical damage, there remain several challenges. One challenge is whether the proposed deep learning strategies could outperform the conventional physics-based methods or shallow learning for signals after coupled scattering and reflection/refraction with high levels of noise. 

Another challenge is whether the trained model could maintain high accuracy when the labeled data are “contaminated” by structural uncertainty, for instance, various operational and environmental conditions. Please note that the performance of the detectability using the deep learning could be fragile, when handling unlabeled damage state data, particularly outside of representation. Therefore, we will not address this in this study.

Therefore, the attempts below were made to further elucidate the effectiveness of the proposed damage detection methods as compared to conventional methods, and determined its robustness when handling the data including structural uncertainty. 

### 6.1. Effectiveness of the Deep Learning Used in This Study for the Guided Wave Signal Process as Compared to Physics-Based or Shallow Learning Methods 

In this section, we attempted to demonstrate the effectiveness of the proposed CNN-based damage detection by comparing it with physics-based and shallow learning methods. For simplicity, we used Case 3 in Table 1 and the RMS (in Equation (2) in Section 2.3) was selected as the physics-based method and SVM, one of the best shallow learning strategies, was used to demonstrate the concept, and the results were plotted in Figure 28. 

As clearly illustrated in Figure 28, RMS performed poorly at all cases, suggesting that the physics-based method had low detectability for the damage states under even low noise levels. The accuracy of the prediction using the physics-based method dramatically dropped with the increase of the noise level, even under the SNR of 90 dB, which often had no effects on the effectiveness of the deep learning as observed in Section 5 and in Figure 28. 

Figure 28 revealed that both shallow learning and deep learning-based methods demonstrated high accuracy for most cases, with almost 100% accuracy, even at the noise level of SNR of 70 dB. Similar to previous observation, when the noise level increased to the SNR of 60 dB, both methods had a sharp drop in accuracy due to high noise interference to signals. The performance of the deep learning stayed 76% accuracy, by 25.3% higher accuracy over that of the SVM. 

### 6.2. Robustness of the Pre-Trained Deep Learning in This Study via Blind Test

Pipelines often experience different operational conditions. For instance, underground pipelines are usually embedded under different types of soils or cast in concrete. The propagation of the guided waves in pipelines could reflect and decay soon due to leakage of energy radiating into the embedding material and energy loss absorbed by the embedding material. As a result, structural uncertainty due to attenuation from various embedding materials poses more complexity for guided wave signal processing. Therefore, we explored Case 4 with structural uncertainty using soil or concrete as the embedding medium in this section to demonstrate the robustness of the methods. As illustrated in Figure 29, the new dataset was generated in COMSOL, where the pipes with different severities of welding defect as used in Case 4 were embedded in soil or concrete, with a depth of 0.5 m. The pipe size and its properties were identical to those in Section 4, while the soil used in the model had properties of: density of 2600 kg/m^3^, Young’s modulus of 20 MPa, and Poisson’s ratio of 0.2, and concrete was: density of 2500 kg/m^3^, Young’s modulus of 32.5 GPa, and Poisson’s ratio of 0.16. 

Instead of updating the data classifier we trained in Section 4 using the new data, we used the new dataset as a blind test to validate the robustness of the pre-trained damage detection method, i.e., extrapolation or prediction capability for handling new data that are outside of the training datasets. As stated, these models were originally derived from the extension of Case 4 with a certain level of structural uncertainty, and thus the data generated were still compatible within a certain level of confidence. 

Results were plotted in Figure 30 and Figure 31, where pipes without embedment were extracted from Section 4 as the reference for a comparison to other embedment cases. 

Figure 30 and Figure 31 revealed that the embedding materials reduced the accuracy in detectability, mainly due to more complex waves from attenuation (energy loss) and reflections of boundaries, particularly sensitive to noise interference. As illustrated in Figure 31, the pre-trained model could better predict the results of the case with the embedding soil with much higher accuracy, as compared to that of the embedded concrete. It was mainly because the concrete had much higher attenuation that stems from the higher energy leakage of the guided waves to the embedding concrete and energy loss absorbed by concrete as compared to soil, as observed in the literature [50]. 

As illustrated in Figure 30 and Figure 31, the pre-trained model could maintain 100% accuracy to predict the results for the cases with the embedding soil under the noise level of SNR of 80 dB. However, the classification dropped to 83.3% when the noise level approached to SNR of 70 dB. Close look revealed that 19 signals with weld defect of 10% severity were mislabeled to the 5% severity, while 13 data point of the case of 5% severity are misclassified as the 1% severity. More mislabeling could be found when the noise level was 60 dB that led to low accuracy by 60.3%. 

Figure 30 and Figure 31 show that the embedding concrete demonstrated much lower accuracy in prediction, particularly with the increase of noise interference. The accuracy was 97.3% at the noise level of SNR = 100 dB, including eight data points misclassified into the wrong labels. The accuracy dropped nearly linearly with the increase of the noise levels. As illustrated in Figure 30 and Figure 31, when the noise level approached to SNR of 70 dB, the detectability reduced to 42% accuracy, which was much lower than that of the embedded soil, suggesting that fully embedding concrete could significantly posed a challenge for the guided waves used for damage state classification. 

## 7. Conclusions

This study investigated deep learning-enriched damage detection of metallic pipes with weldments using ultrasonic guided waves. Welding defects and material discontinued damages were detected by the ultrasonic guided waves and identified by eight-layer CNN models with high accuracy. The dataset of ultrasonic guided waves traveled through the pipeline was generated by the simulation. The detection results were associated with classification classes in which each class represented a condition scenario of the pipeline structures, including damage types, positions, and severities. Damage interaction was considered in this study. The effectiveness of the methods interference by noise was examined. Some conclusions can be drawn as follows:(a)The proposed deep learning networks showed high detectability and high accuracy for signal process of ultrasonic guided waves and automatically extracting the sensitivity features by appropriate filter sizes and parameters.(b)The results demonstrated that the deep learning model could be effective tools for the data classification of interacting threats by combined effects of damages and weldment, and extract sensitive information for localization of the notch-shaped damage in the pipeline with weldment. The model showed high performance with 100% accuracy, even when the noise level was as high as SNR of 80 dB. When the noise level reached up to 60 dB, more misclassifications were observed.(c)Results further confirmed that the proposed deep learning-based damage detection could have high effectiveness, when welding defects and notch-shaped damage appeared simultaneously. The proposed data classifier could still maintain high accuracy for detectability, especially when the noise level is 70 dB or lower.(d)The proposed data classifier was also effective to classify the weld defect types and severities at high noise levels (even at SNR of 70 dB). The accuracy of the result dropped to around 70% when SNR was 60 dB, but the results still outperformed well over conventional physics-based and shallow learning methods(e)Further blind test revealed that the proposed methods could ensure the high accuracy and the robustness to handle new dataset with structural uncertainty.(f)The limited cases presented in this study may not provide a broader diversity for data representation. Thus, the improvement of the data diversity could further help to verify and calibrate the effectiveness of the proposed deep learning for practical applications in field.

## Figures and Tables

**Figure 1 sensors-22-05390-f001:**
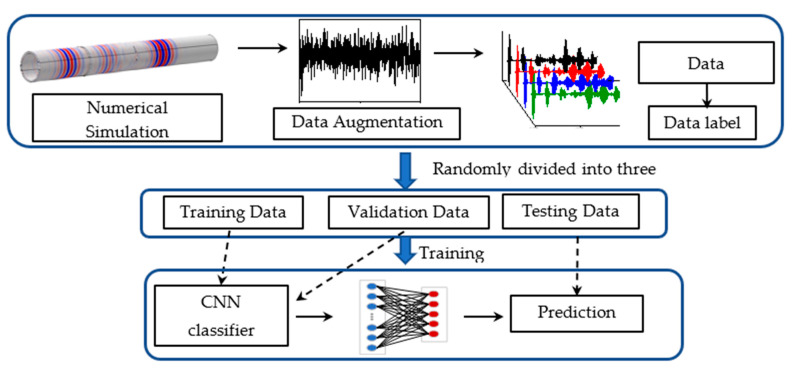
Flow of the proposed concept in this study.

**Figure 2 sensors-22-05390-f002:**
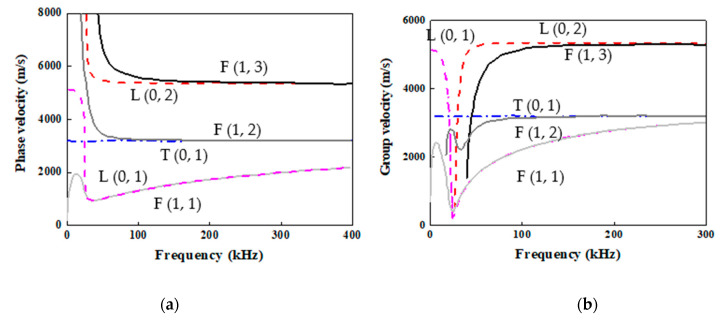
Phase velocity and group velocity. (**a**) Phase velocity. (**b**) Group velocity.

**Figure 3 sensors-22-05390-f003:**
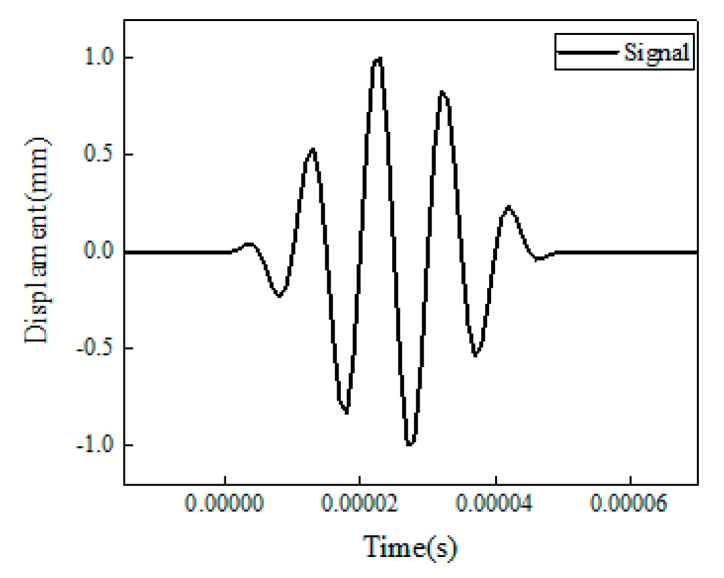
Excited guided waves used in the simulation.

**Figure 4 sensors-22-05390-f004:**
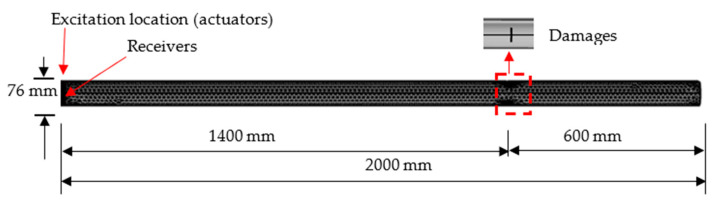
Pipe model from the literature.

**Figure 5 sensors-22-05390-f005:**
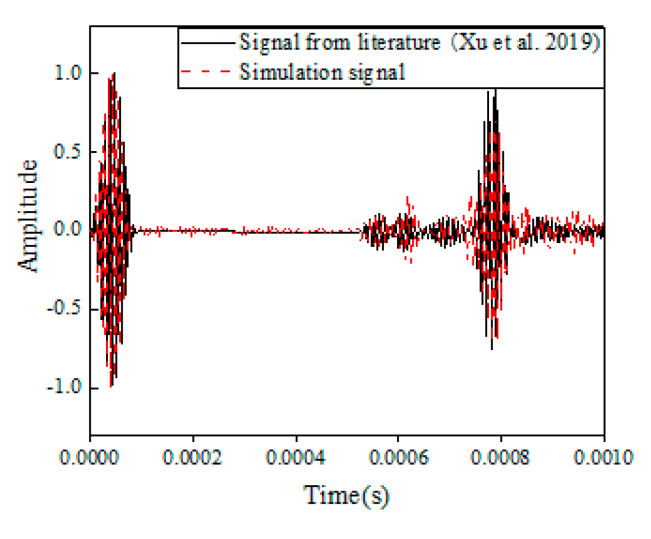
Comparison of the results to those from the literature [44].

**Figure 6 sensors-22-05390-f006:**
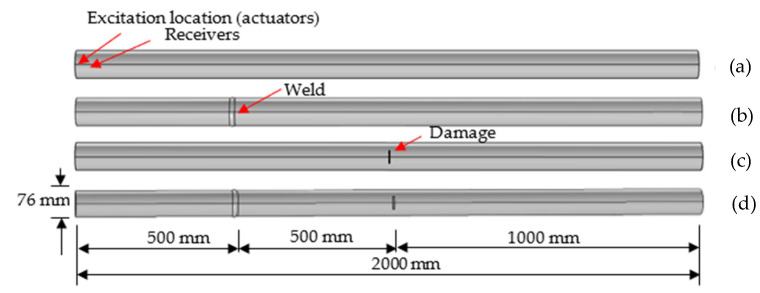
COMSOL models of pipes in different states: (**a**) intact pipe; (**b**) pipe with a weld; (**c**) pipe with a crack; (**d**) pipe with crack and weld.

**Figure 7 sensors-22-05390-f007:**
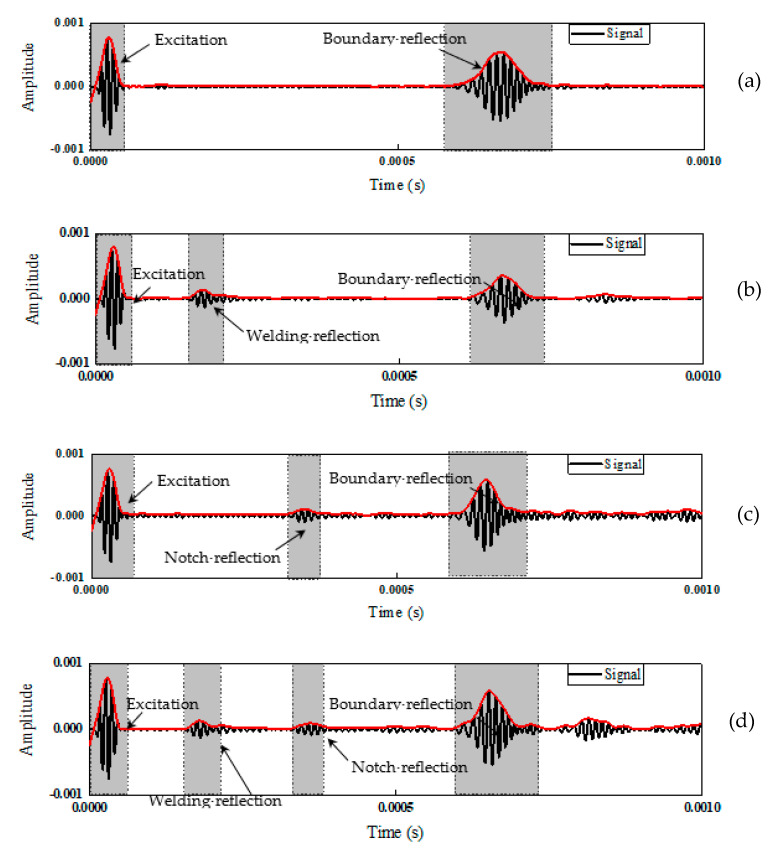
Signal characteristics of the pipes under varying cases: (**a**) intact pipe; (**b**) pipe with a weld; (**c**) pipe with a crack; (**d**) pipe with crack and weld.

**Figure 8 sensors-22-05390-f008:**
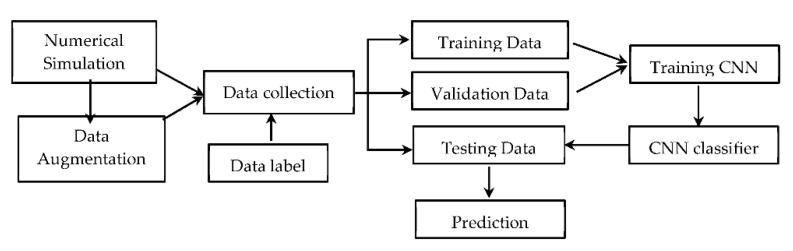
Flowchart for damage detection using the CNN.

**Figure 9 sensors-22-05390-f009:**
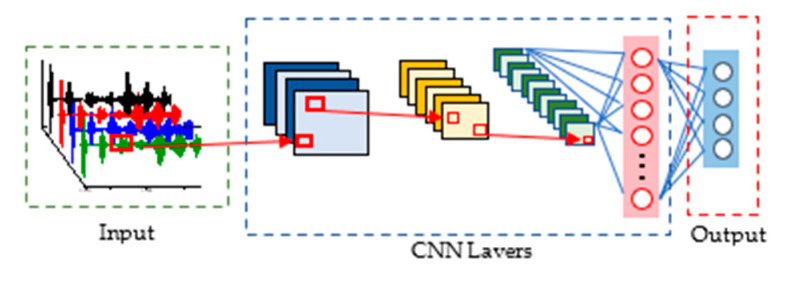
Flowchart for the damage detection by CNN.

**Figure 10 sensors-22-05390-f010:**
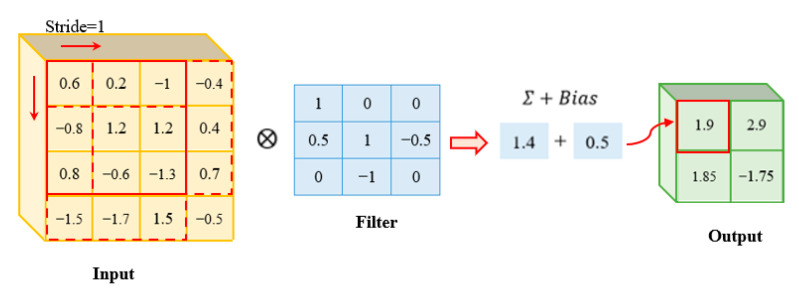
Convolutional layer.

**Figure 11 sensors-22-05390-f011:**
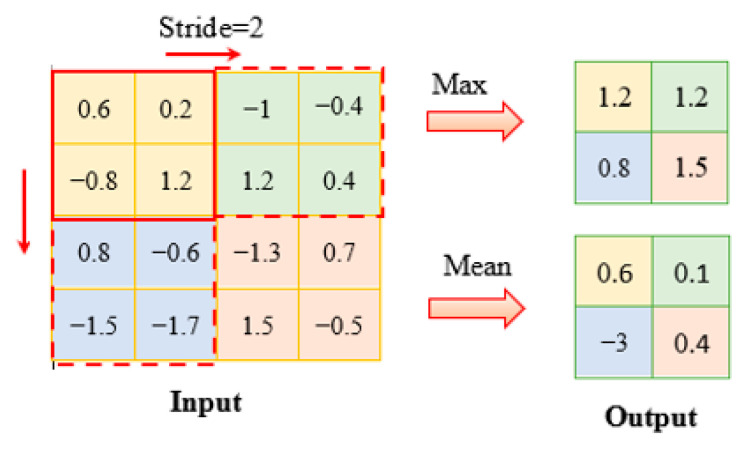
Pooling layer.

**Figure 12 sensors-22-05390-f012:**
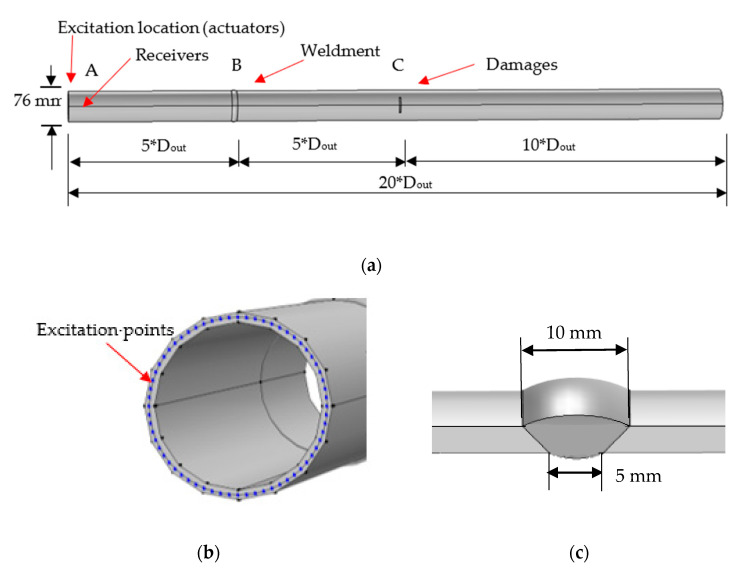
FE modeling of a pipe. (**a**) Steel pipe with a welded joint and notch-shaped damage. (**b**) Excitation nodes. (**c**) V-shaped weldment.

**Figure 13 sensors-22-05390-f013:**

Welding defects (**a**) Lack of fusion; (**b**) Cracks; (**c**) Undercut; (**d**) Lack of penetration.

**Figure 14 sensors-22-05390-f014:**
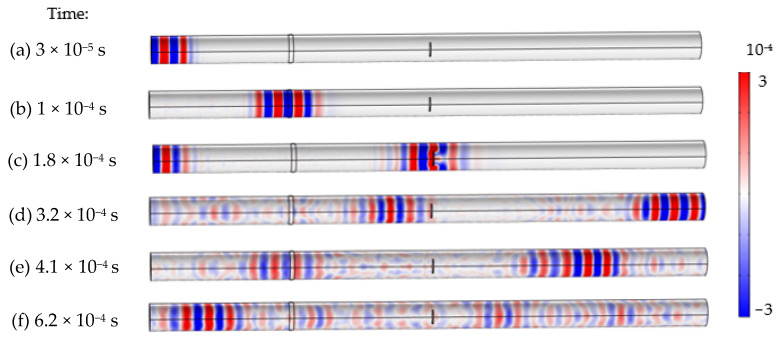
Wave propagation through the entire span of the pipe: (**a**–**f**).

**Figure 15 sensors-22-05390-f015:**
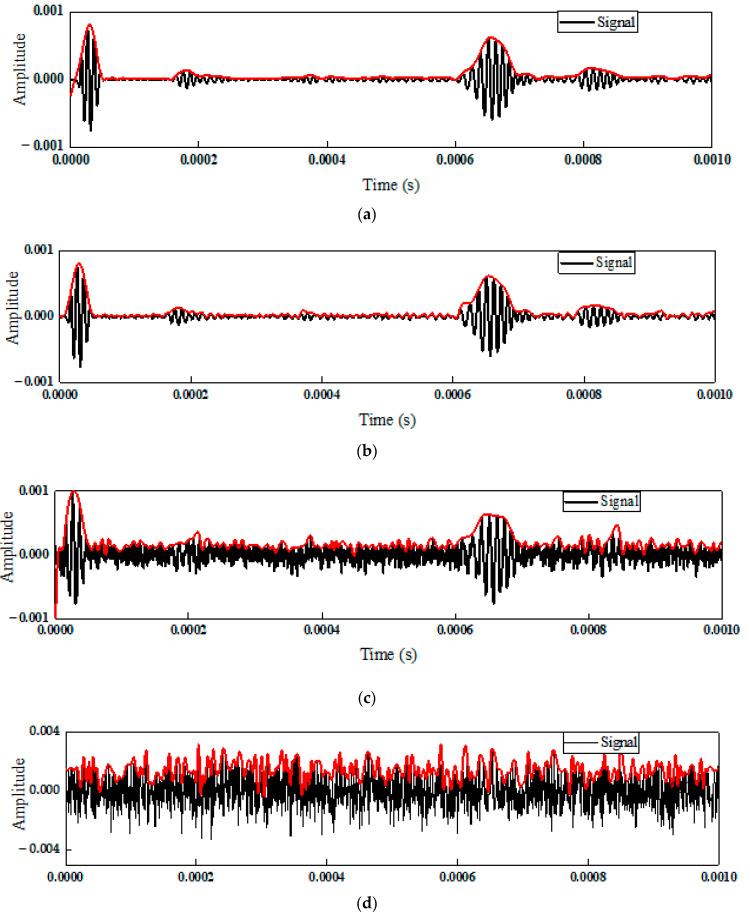
Signals with different noise levels. (**a**) Original signal; (**b**) SNR = 100 dB; (**c**) SNR = 80 dB; (**d**) SNR = 60 dB.

**Figure 16 sensors-22-05390-f016:**
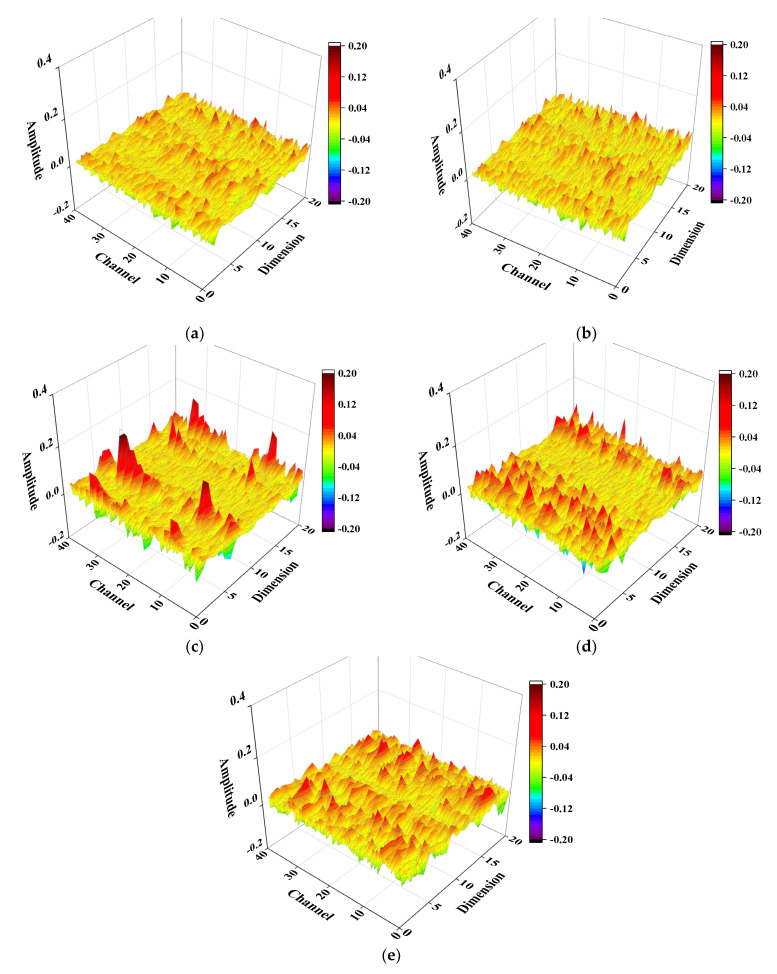
Weights of the third convolutional layer at 60 dB: (**a**) Group 1; (**b**) Group 2; (**c**) Group 3; (**d**) Group 4; (**e**) Group 5.

**Figure 17 sensors-22-05390-f017:**
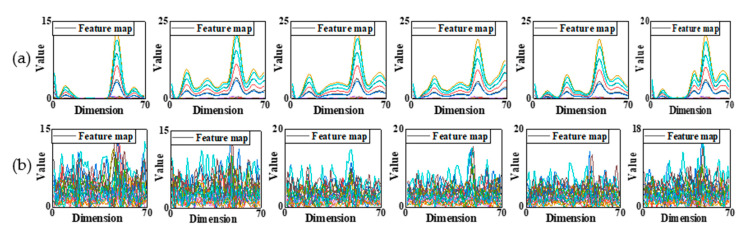
Feature maps. (**a**) SNR = 100 dB; (**b**) SNR = 60 dB.

**Figure 18 sensors-22-05390-f018:**
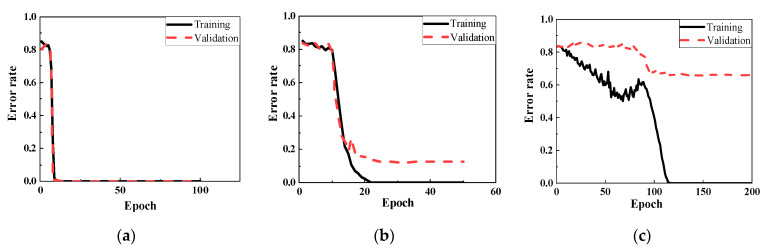
Training and validation results under noise of. (**a**) SNR = 80 dB; (**b**) SNR = 70 dB; (**c**) SNR = 60 dB.

**Figure 19 sensors-22-05390-f019:**
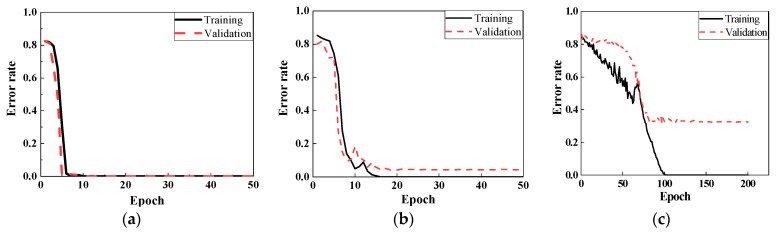
Training and validation results. (**a**) SNR = 80 dB; (**b**) SNR = 70 dB; (**c**) SNR = 60 dB.

**Figure 20 sensors-22-05390-f020:**
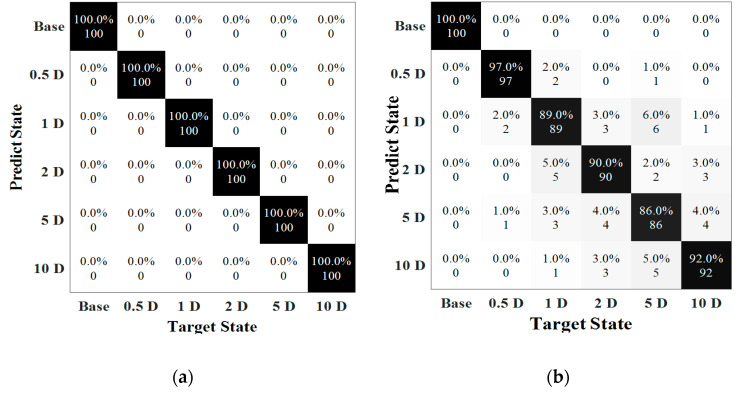
Testing results. (**a**) SNR = 80 dB (Accuracy = 100%); (**b**) SNR = 70 dB (Accuracy = 92.3%).

**Figure 21 sensors-22-05390-f021:**
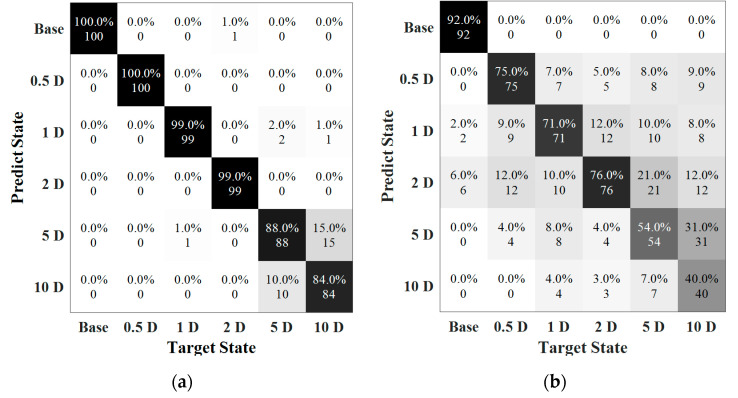
Testing results. (**a**) SNR = 70 dB (Accuracy = 95%). (**b**) SNR = 60 dB (Accuracy = 68%).

**Figure 22 sensors-22-05390-f022:**
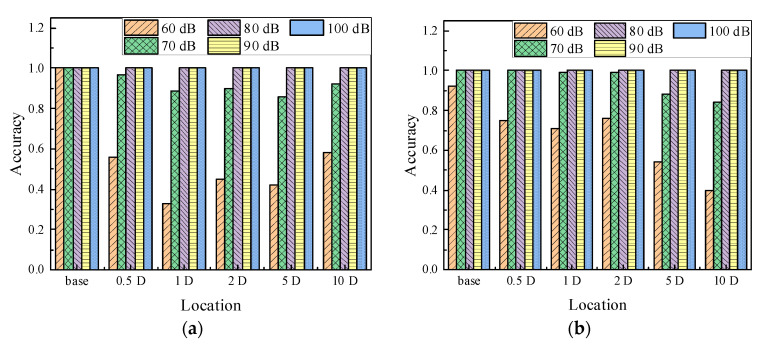
Locations identification in pipeline. (**a**) Location detects without welding defect. (**b**) Location detects with welding defect.

**Figure 23 sensors-22-05390-f023:**
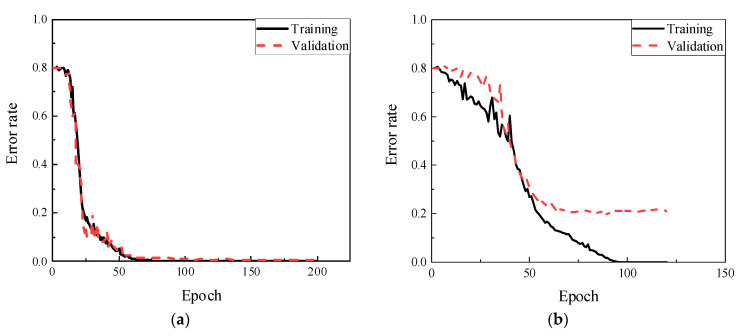
Training and validation results. (**a**) SNR = 70 dB. (**b**) SNR = 60 dB.

**Figure 24 sensors-22-05390-f024:**
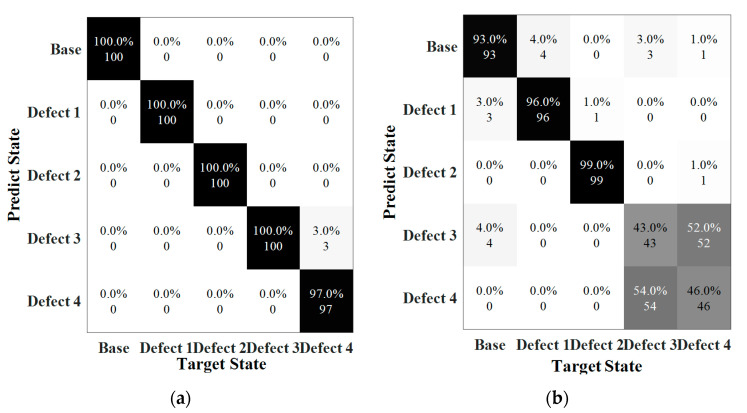
Testing results. (**a**) SNR = 70 dB (Accuracy = 99.4%). (**b**) SNR = 60 dB (Accuracy = 75.4%).

**Figure 25 sensors-22-05390-f025:**
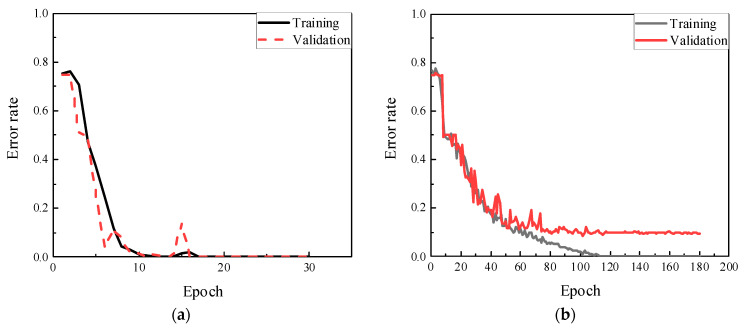
Training and validation results. (**a**) SNR = 80 dB (**b**) SNR = 70 dB.

**Figure 26 sensors-22-05390-f026:**
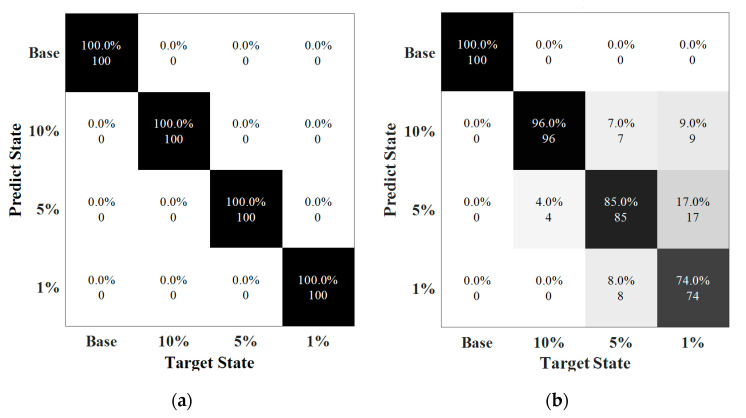
Confusion matrix associated with defect severity. (**a**) SNR = 80 dB (Accuracy = 100%). (**b**) SNR = 70 dB (Accuracy = 88.8%).

**Figure 27 sensors-22-05390-f027:**
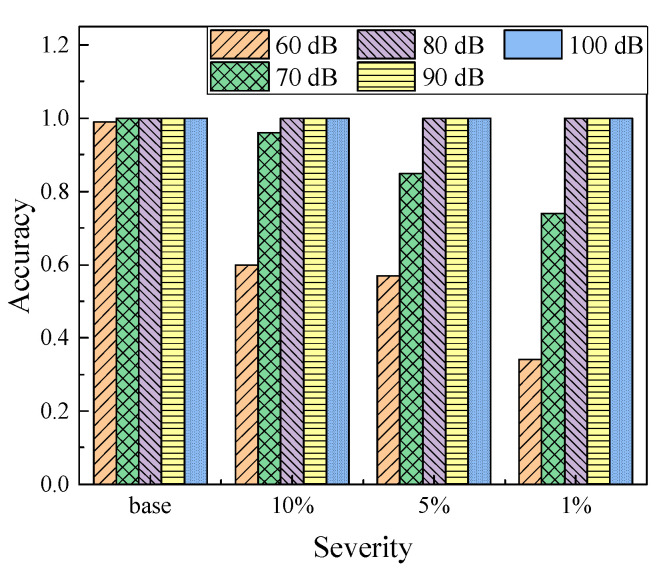
Accuracy with respect to defect severity under various noise levels.

**Figure 28 sensors-22-05390-f028:**
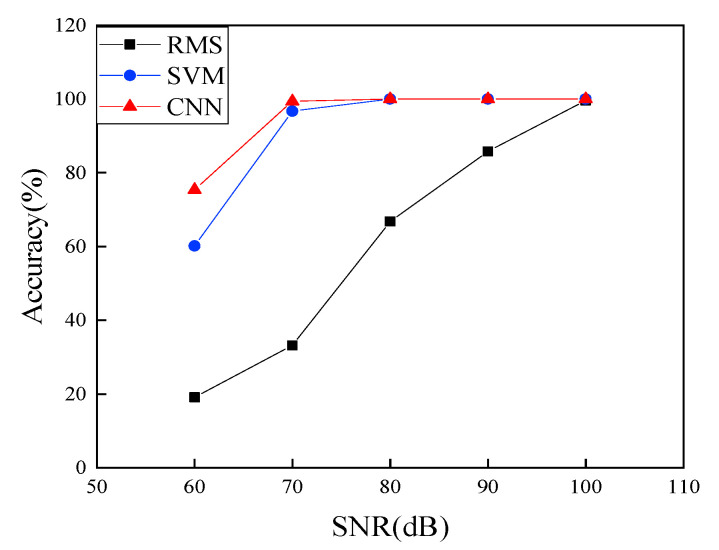
Comparison of the proposed damage detection with other two methods in accuracy.

**Figure 29 sensors-22-05390-f029:**
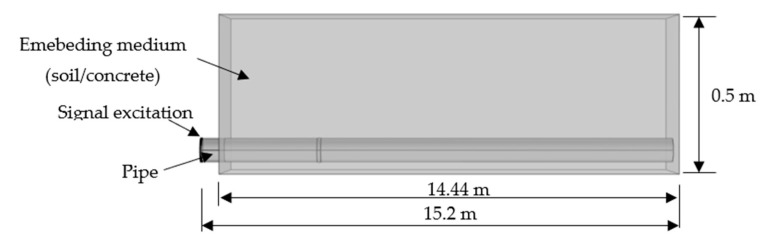
Model of a buried pipe.

**Figure 30 sensors-22-05390-f030:**
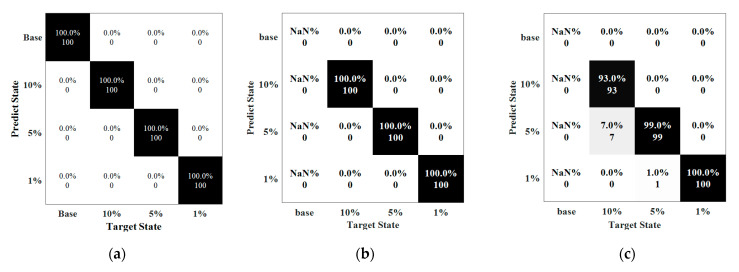
Detectability for pipes under different embedment conditions: (**a**–**f**). (**a**) Pipe without embedment (Accuracy = 100%, SNR = 100 dB). (**b**) Pipe with embedding soil (Accuracy = 100%, SNR=100 dB). (**c**) Pipe with embedding concrete (Accuracy = 97.3%, SNR = 100 dB). (**d**) Pipe without embedment (Accuracy = 88.8%, SNR = 70 dB); (**e**) Pipe with embedding soil (Accuracy = 82%, SNR = 70 dB); (**f**) Pipe with embedding concrete (Accuracy = 42%, SNR = 70 dB).

**Figure 31 sensors-22-05390-f031:**
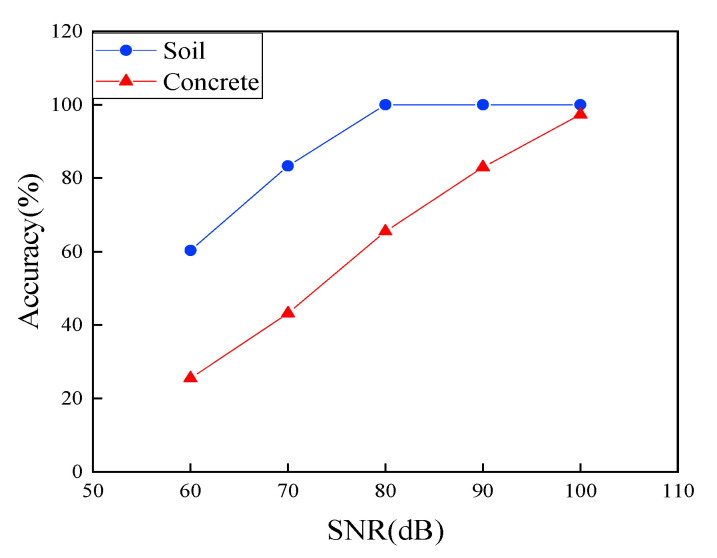
Accuracy of the models for embedding cases with respect to different noise levels.

**Table 1 sensors-22-05390-t001:** Test matrix for computation modeling.

Case	Label	Damage Location	Damage Size	Damage Depth	Welding Defect Type	Severity of Welding Defect	Noise Interference
Reference	State #1	/	/	/	/	/	Noise levels of from 60 dB to 100 dB
Case 1: Variance due to damage location	State #2	0.5 × D_out_	0.1 × D_out_	4 mm	/	/	
	State #3	1 × D_out_	0.1 × D_out_	4 mm	/	/	
	State #4	2 × D_out_	0.1 × D_out_	4 mm	/	/	
	State #5	5 × D_out_	0.1 × D_out_	4 mm	/	/	
	State #6	10 × D_out_	0.1 × D_out_	4 mm	/	/	
Case 2: Variance due to damage location with weld defect	State #7	0.5 × D_out_	0.1 × D_out_	4 mm	Defect 1	1%	
	State #8	1 × D_out_	0.1 × D_out_	4 mm	Defect 1	1%	
	State #9	2 × D_out_	0.1 × D_out_	4 mm	Defect 1	1%	
	State #10	5 × D_out_	0.1 × D_out_	4 mm	Defect 1	1%	
	State #11	10 × D_out_	0.1 × D_out_	4 mm	Defect 1	1%	
Case 3: Variance due to the type of weld defect	State #10	5 × D_out_	0.1 × D_out_	4 mm	Defect 1	1%	
	State #12	5 × D_out_	0.1 × D_out_	4 mm	Defect 2	1%	
	State #13	5 × D_out_	0.1 × D_out_	4 mm	Defect 3	1%	
	State #14	5 × D_out_	0.1 × D_out_	4 mm	Defect 4	1%	
Case 4: Variance due to severity of weld detect	State #10	5 × D_out_	0.1 × D_out_	4 mm	Defect 1	1%	
	State #15	5 × D_out_	0.1 × D_out_	4 mm	Defect 1	5%	
	State #16	5 × D_out_	0.1 × D_out_	4 mm	Defect 1	10%	

**Table 2 sensors-22-05390-t002:** Results of different filter sizes.

	Filter Size of the Convolutional Layer			Testing		
	1st	2nd	3rd	(70 dB)	(65 dB)	(60 dB)
Group 1	5 × 2	5 × 3	5 × 1	100%	95.33%	61.00%
Group 2	15 × 2	15 × 3	5 × 1	100%	94.83%	75.33%
Group 3	25 × 2	25 × 3	5 × 1	100%	99.17%	82.83%
Group 4	35 × 2	35 × 3	5 × 1	100%	98.67%	78.17%
Group 5	45 × 2	45 × 3	5 × 1	100%	97.67%	79.83%

**Table 3 sensors-22-05390-t003:** Results of different filter numbers.

	Number of the Convolutional Layer			Testing		
	1st	2nd	3rd	(70 dB)	(65 dB)	(60 dB)
Group 1	10	20	10	100%	96.67%	74.83%
Group 2	15	30	15	100%	92.17%	73.50%
Group 3	20	40	20	100%	99.17%	82.83%
Group 4	25	50	25	100%	98.00%	81.50%
Group 5	30	60	30	100%	98.00%	78.67%

**Table 4 sensors-22-05390-t004:** Results of different Pooling sizes.

	Size of the First Pooling Layer	Size of the Second Pooling Layer	Type of Pooling Layer	Testing (65 db)
Group 1	10	5	max	98.67%
Group 2	10	5	average	91.00%
Group 3	15	8	max	99.17%
Group 4	15	8	average	98.00%
Group 5	20	10	max	93.5%
Group 6	20	10	average	68.5%

**Table 5 sensors-22-05390-t005:** Detail of the proposed CNN.

Name	Filters	Filter Size	Stride	Bias	Output Layer Size
Input layer	--	--	--	--	2000 × 4
Convolutional layer (C_1_)	20	25 × 2	1	20	1976 × 3
Max pooling layer (P_1_)	20	15 × 1	5	--	393 × 3
Convolutional layer (C_2_)	40	25 × 3	1	40	369 × 1
Max pooling layer (P_1_)	40	8 × 1	5	--	73 × 1
Convolutional layer (C_3_)	20	5 × 1	1	20	69 × 1
ReLU	--	--	--	--	69 × 1
Full connected layer (F_1_)	4	69 × 1	1	4	4
Softmax	--	--	--	--	4

## Data Availability

The data presented in this study are available on request from the corresponding author.

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
