# Peer review of "Deep Learning Empowered Structural Health Monitoring and Damage Diagnostics for Structures with Weldment via Decoding Ultrasonic Guided Wave"

_sensors, 2022, doi:10.3390/s22145390_

Round 1
Reviewer 1 Report
This work is very interesting. I would suggest the author to check again the whole paper cause although in most of the paper the language is fine, in many spots it is a bit “strange” (e.g. “The prediction was 100% correctly in the certain noise levels, 80dB or higher. With the noise level increased to 640 60 dB, more misclassifications were appeared”)
Moreover in line 114 “piezoelectric actuators” I would suggest to be changed with “piezoelectric transducers ”
Also I would like from the authors to clarify when the results become totally untrustworthy “The prediction was 100% correctly in the certain noise levels, 80dB or higher. With the noise level increased to 640 60 dB, more misclassifications were appeared”. I would suggest that it should be mention till what dB the results are acceptable I can see that many times different dBs are mentioned but even the 85 p% accuracy is acceptable? Also did the authors benchmarked the results with other common NDE techniques.
Moreover does the deep learning and the AI be “upgraded” to include reslts from other techniques in order to have a higher accuracy?
Also in figure 13 please at (d) at the 4th image
Reviewer 2 Report
A deep learning method, convolutional neural network (CNN), was developed for characterizing welding defect in pipelines by processing ultrasonic guided wave signals under various signal to noise ratio and structural uncertainties. And its effectiveness for damage classification was proved. There are some questions should be revised:
1. The principle of the developed method for ‘decoding ultrasonic guided waves’ should be clearly descripted.
2. The SNR of signals can be easily improved with signal processing methods, while echoes from various welding defect results wave mode overlapping, which has serious affect the accuracy interpretation of ultrasonic guided waves. Does the developed method still have reliable performance under the case?
3. Some intelligent algorithms besides the ANN used for guided waves processing should be reviewed in the introduction, and following papers could be considered by the authors
Deep learning analysis of ultrasonic guided waves for cortical bone characterization. IEEE Transactions on Ultrasonics, Ferroelectrics, and Frequency Control, 2020, 68(4): 935-951.
Evolutionary strategy-based location algorithm for high-resolution Lamb wave defect detection with sparse array. IEEE Transactions on Ultrasonics, Ferroelectrics, and Frequency Control. 2021, 68(6): 2277-2293.
4. The figures should be drawn clearly, e.g., the title, description, and units of axis. For example, the title of Fig. 2 was missing, and the curves of L(0,1) and L(0,2) should be different in display. The ‘phase/ group speed' is recommended revised as ‘phase/ group velocity’.
5. Please discuss the reasons of the low accuracy of CNN classifier under the low SNR.
6. "2.1. Dispersion Curve " should be 2.1. Dispersion Curves and in some places guided wave should be guided waves.
Reviewer 3 Report
The paper discussed using deep learning (convolutional neural network) for classifying guided wave signals on weldment defects and defect locations. The motivation for using guided wave over conventional weld UT methods is unclear. The way to generate simulation data in this work is not reasonable, and the data has limited diversity. Detailed comments are listed below:
1. A COMOSOL model of the pipe should be provided in Section 2.2
2. COMCOL models with girth weldment and notch should be provided at least for Figure 5(b)-(d). Otherwise, it will be confusing how the ultrasonic signals in Figure 5 are generated.
3. L311-L312: It should be: Notch-shape damage at point C and weldment at point B?
4. Is 0.4 too large for the assumed Poisson’s ratio of Ti-6Al-4V? According to the literature, it is around 0.35.
5. Does Group #1-#5 represent the five notch locations (0.5Dout to 10 Dout)? If yes, this should be clarified in Section 4.
6. State #15 is missing in Table 1.
7. The different levels of severity in table 1 should be clarified in Section 4.1.
8. Classification of notch location for case 1 and case 2 does not contain too much significance since this can be achieved with conventional guided wave techniques. The only difficulty here is the existence of the weldment. However, the signal packets of the weldment and notch can be easily separated in the time-domain signal.
9. The training and testing data come from the same ultrasonic signal with white noises. This cannot guarantee the diversity of the data. As a result, the classification performance will be very good. However, this can not validate the use of CNN. A reasonable method to generate the simulation data is to tweak the defect positions and sizes for each signal.
1 10. The reviewer suggests the work should focus on cases 3 and case 4. The effect of signal noise level on the model performance is also an interesting aspect. The advantage of using guided wave for welding defect inspection is not clearly discussed. While the convention weld inspection methods (e.g., phase array) focus on the local area around the weld, the guided wave method is more easily affected by other pipe defects or discontinuities and becomes less sensitive to weld defects.
Round 2
Reviewer 3 Report
The authors have clearly addressed the comments of the reviewers. The reviewer does not have further comments and the paper can be accepted in its present form.